# Swedish private-sector employees' experiences of promoting and hindering factors for working while having mental health problems: A qualitative study

Anna Frantz[1]*, Iben Axén[1], Gunnar Bergström[1,2], Anna Finnes[3,4], Elisabeth Björk Brämberg[1,5]

1 Unit of Intervention and Implementation Research for Worker Health, Institute of Environmental Medicine, Karolinska Institutet, Stockholm, Sweden, 2 Department of Occupational Health, Psychology and Sports Sciences, Faculty of Health and Occupational Studies, University of Gävle, Gävle, Sweden, 3 Division of Insurance Medicine, Department of Clinical Neuroscience, Karolinska Institutet, Stockholm, Sweden, 4 Academic Primary Healthcare Centre, Region Stockholm, Stockholm, Sweden, 5 School of Public Health and Community Medicine, Institute of Medicine, University of Gothenburg, Gothenburg, Sweden

* anna.frantz@ki.se

## Abstract

### Objectives

The workplace is an important arena for identifying and managing early symptoms of mental health problems. This study aimed to explore how private sector employees at risk of sickness absence due to mental health problems experienced promoting and hindering factors for working.

### Methods

Semi-structured interviews were conducted with 18 employees with mental health problems (≥3 on the General Health Questionnaire or who answered yes to a question on self-predicted sickness absence in the coming year due to common mental disorders). Reflexive thematic analysis was used to analyze the data.

### Results

The analysis resulted in three themes: influence of life stage on working while having mental health problems; managing mental health problems in the social and organizational context; and preserving one's identity and agency when working while having mental health problems. Promoting and hindering factors occurred at the individual, group, leader and organizational levels. Promoting factors included social support from colleagues, a trusting relationship with the first-line manager, and engaging in leisure-time physical activity. Hindering factors included perceiving the cause of

**Data availability statement:** The data are not publicly available as they include information that could compromise the privacy of the study participants (data contain potentially identifying and sensitive study participant information). Reasonable inquiries about access may be sent to Karolinska Institutet, Institute of Environmental Medicine, Unit of Intervention and Implementation Research for Worker Health, Box 210, 171 77 Stockholm or by contacting the Research and Data Office at Karolinska Institutet: rdo@ki.se. The Swedish Ethical Review Authority will then be contacted for permission.

**Funding:** EBB no. 190134 AFA Insurance https://fou.afaforsakring.se/sv The funding body had no role in the study design; data collection and analysis; interpretation of data and results; writing of the paper; or decision to submit for publication.

**Competing interests:** The authors have declared that no competing interests exist.

**Abbreviations:** HAD, Hospital anxiety and depression scale; IGLOO, Individual, group, leader, organization, overarching context; JD-R, Job demands resources model; MHP, Mental health problems; RCT, Randomized controlled trial.

symptoms to be primarily outside work leading to delayed access to help, a cold or noisy work environment, and schedule mismatch.

## Conclusion

Promoting and hindering factors occurred at multiple levels. Fostering a supportive and inclusive work environment where there is space for enjoyment can help employees manage mental health problems at work. Other promoting factors include having an active and present manager, reasonable production goals, and leisure-time physical activity. Employees who perceive the cause of their symptoms to be related to their private life tend not to seek help from the workplace, despite the impact on their work. This should be considered in the development of future interventions. Our study supports a life-course perspective on the understanding of how employees experience promoting and hindering factors for working while having mental health problems.

## Introduction

Mental health problems (MHPs, including anxiety, depression, stress-related disorders and sub-threshold conditions [1]) affect nearly one in five of the working-age population in the Organisation for Economic Co-operation and Development member states [2]. Most of those affected are working despite being more prone to spells of sickness absence compared with those with other chronic conditions [3], making the workplace an important arena for early management of MHPs, (e.g., identification of early symptoms, provision of support) [2]. Working while having MHPs can be conceptualized through presenteeism – the behavior of going to work while experiencing ill-health [4]. Previous research has studied the potential negative impact of presenteeism on work productivity and employee health [4,5]. However, presenteeism has also been suggested to include potential functional or therapeutic aspects, such as being an individual strategy to maintain or regain function [6]. In this study, in line with this conceptualization, working while having mental health problems (presenteeism) is therefore seen as a behavior that may have negative or positive implications [4].

Factors in the work environment may contribute to improvement of MHPs (e.g., supportive supervision [7]), or deterioration of MHPs (e.g., lack of decision latitude [8,9]). From an organizational perspective, several job stress theories describe how work may affect individuals and be the precursor of a job situation that fosters motivation and health, or a job situation that ultimately will result in ill-health. One such framework is the Job Demands-Resources (JD-R) model, which applies across different occupations [10]. The JD-R model proposes two processes that affect well-being and job engagement: a health impairment process and a motivational process, providing an understanding of how resources and demands can affect working [11]. The relationship between workplace and private life factors should be explored to understand how employees manage working while

having MHPs. With this in mind, qualitative studies may provide a rich and nuanced understanding of such relationship [12]. Only a few studies have explored experiences of working while having MHPs [13]. A previous study found that employees used a variety of strategies at an individual level to continue working, such as putting on a façade or escaping distressing emotions through social, work or health habits [14]. Employees with MHPs have described distancing themselves from work [15,16], which is not always beneficial to the employee because it can be both protective and isolating [16].

In Sweden, employers have the responsibility to systematically work to reduce risks that can lead to work-related injury or ill-health, including MHPs [17]. Hence, the workplace provides an arena for preventing sickness absence due to MHPs [18] and understanding what promotes or hinders working while having MHPs could elucidate not only individual strategies, but also factors at a group, leader, or organizational level. The aim of the study was to explore how private-sector employees at risk of sickness absence due to MHPs experienced promoting and hindering factors for working while having an MHP.

## Methods

This qualitative study was inspired by constructivist epistemology. Data were collected through semi-structured interviews; the data were seen as co-constructed between the interviewer and the participant. The research group consisted of a multidisciplinary team with experience in qualitative research, as well as research on MHPs and the psychosocial work environment. Data were analyzed inductively using reflexive thematic analysis [19,20] to understand how participants made sense of promoting and hindering factors for working while experiencing MHPs. The IGLOO framework (acronym representing the levels individual, group, leader, organization, and overarching context) [21] was used to indicate the level for each promoting and hindering factor. We have appraised and aimed to follow the guidelines for reporting reflexive thematic analysis in this article [22].

### Participants and procedure

Participants were recruited from a randomized controlled trial (RCT) evaluating a problem-solving intervention provided by first-line managers at three private companies in Sweden [23]. The three companies were large enterprises, covering production, warehouse logistics, and retail. Employees at risk of sickness absence due to MHPs were identified through self-reports scoring with a cut-off of ≥3 points on the General Health Questionnaire 12-item (Swedish version) validated in the general population to detect depression [24], or answering"Yes, most likely" or"Yes, quite likely" to a question on self-rated risk of sickness absence due to MHPs in the year to come (i.e., "About your health – do you think you will receive sick leave benefits because of stress, anxiety or depression in the coming 12 months?"). The response options were "Yes, most likely", "Yes, quite likely", "I'm not sure", "No, probably not". Further inclusion criteria were 18–59 years of age, ability to understand written and spoken Swedish. Exclusion criteria were ongoing leave due to sickness or other reason, pregnancy, sickness absence due to a common mental disorder for ≥14 calendar days during the last 3 months, exposure to workplace bullying by the first-line manager; or planned long-term absence in the coming year [23]. The participants in the study all had jobs that could not be performed outside the workplace. Ethical approval was granted from the Swedish Ethical Review Authority (reference numbers 2020−03114, 2021−01748). The study followed the ethical principles of the Declaration of Helsinki for research involving human subjects [25].

Eligible participants received information about the study and the voluntary nature of participation, before consenting to take part in the study. Consent was given electronically. For the current study, all participants from the RCT were eligible to take part in an interview. A total of 95 of the 129 eligible participants were contacted by phone 1 year after enrollment in the RCT. Reasons for not contacting the remaining 34 were as follows: more than 18 months from inclusion (n = 26), changed workplace (n = 4) and lost contact details (n = 4). Eighteen participants agreed to an interview (intervention group = 7, control group = 11).

## Data collection

An interview guide was developed by Anna Frantz (AFr), Elisabeth Björk Brämberg (EBB), Iben Axén (IA), and Anna Finnes (Afi) based on previous literature on psychosocial factors and work-life balance [9,26], while Gunnar Bergström (GB) read and approved the final version. The guide included probing questions to help the informants reflect on factors that promoted or hindered working during the previous year. The first draft of the interview guide was tested in two pilot interviews, not included in the dataset. Based on these interviews, AFr and EBB discussed the relevance of the questions and produced a final version of the interview guide. The interview started with a broad, open question: "Tell me about your work. What does a normal working day look like?" Examples of questions to follow were "If you think about the workplace, was there anything that made it easier to stay at work?" and "Did you use any specific strategies to be able to stay at work?". Prompts were used to encourage elaboration, e.g., "Could you give an example of a situation?" or "Could you tell me more about that?". The full interview guide is provided in S1 File.

Interviews were conducted by AFr between January 2023 and August 2023. One participant preferred a video-conference call, the other interviews were conducted by phone. The interviews were audio-recorded and lasted between 21 and 62 minutes. Notes were taken after each interview. These notes were not included in the analysis but were used to support reflexivity.

## Data analysis

Interviews were transcribed verbatim, and the transcripts were checked for accuracy by AFr. The reflexive thematic analysis followed the six phases suggested by Braun and Clarke [19]. Reflexivity during the analysis was facilitated in multiple ways: by notetaking as a way to keep track of initial ideas and reflections, having two researchers coding data, and through continuous meetings with the entire research group to discuss the assumptions around the data and results. In the first phase of the data analysis, AFr listened to each interview and noted her initial thoughts. The transcripts were then read through multiple times by AFr to become familiarized with the data. Notes were taken on initial ideas and reflections for each interview and for the material as a whole. The notes could be a summary of "the story" of the interview, thoughts on what was said, or significant quotes.

The notes were used during the coding procedure in phase two to stimulate ideas and to check if the initial understanding was confirmed or challenged. Inductive coding was conducted in Microsoft Word using the comment function, and coding was organized in Microsoft Excel to keep track of the codes and material. Coding was performed by AFr and EBB. AFr coded the first six interviews independently; then, AFr and EBB coded two interviews together to check for similarities and differences. To enhance reflexivity, differences in coding and assumptions on the meaning of the data were discussed between AFr and EBB. These discussions helped bring assumptions to the surface and broaden the interpretation of the data. The remaining 10 interviews were coded by AFr. Thereafter, all authors met on several occasions to discuss the analysis. During these meetings, the reflective notetaking, thematic mapping, and the progress in writing the report were used to support the discussions and joint data analysis process through checking if previous interpretations of the data were challenged or supported. The meetings continued in the following four phases. In the third phase, Microsoft Excel was used to get an overview and organize the material, identifying patterns in the data and formulating potential themes. In the fourth phase, the codes and themes were reviewed by manually writing the themes on Post-it notes and organizing the codes under the themes to gain an understanding of the clarity of each theme. Fig 1 illustrates the process of coding text extracts related to promoting and hindering factors for working while having MHPs, identifying patterns and organizing codes into themes that convey a common understanding across the data. Although the overview may seem like a linear process from text to codes to themes, this process involved going back and forth through the material, coding extracts with multiple codes before deciding on the code that most clearly answered the research question, and mapping and re-mapping the codes into themes. Through an iterative process of naming themes and writing the report (phases five and six), the themes were reviewed, named, and agreed upon within the author group.

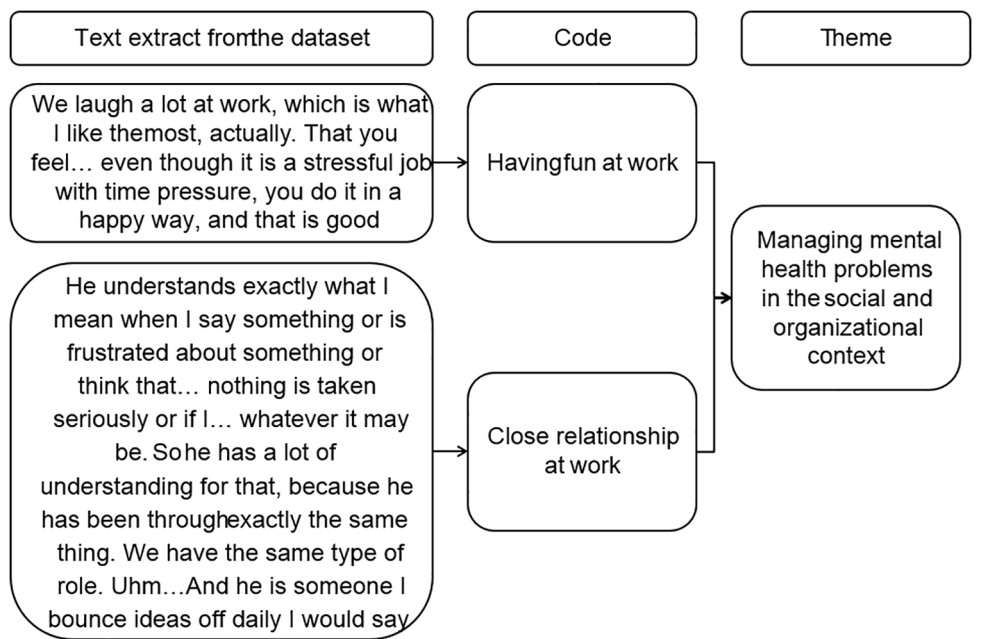

**Fig 1. Example of the analysis process from identifying and coding text extracts to organizing these into themes.**

## Results

An overview of the demographic characteristics of the employees participating in the study is presented in Table 1.

The analysis resulted in three themes, each covering promoting and hindering factors for working while having MHPs. An overview of the promoting and hindering factors is presented in Table 2.

### Influence of life stage on working while having MHPs

Time is a recurrent theme in the participants' stories, as both the individual and the work evolve. A promoting factor was recognizing the temporal state of a stressful situation with high demands both at work and at home. A hindering factor was developing feelings of resignation to work due to unmanaged issues related to MHPs.

Life stage referred to different stages in both the employees' work- and private-life. When work and private life together were a burden, e.g., having small children while working, this was seen as a phase that would pass. Over time, the participants found a way to accept feelings and situations, and to working while having MHPs. On the other hand, if the participants did not feel heard or if they lacked flexibility, this could result in them becoming distant or detached from work. The participants were at different stages in life. They reflected on how they valued and appraised work issues in relation to their motivation to work; from working to gain promotion to working to receive a salary. Life stage had an impact on how promoting and hindering factors were experienced by the worker.

> I can let it go. It always works out somehow. I probably cared a bit more in the beginning when I was appointed to this role than I do now. I've kind of let go of it. I think it's really nice. Because I notice that it works out anyway. (Interview 7)

### Managing MHPs in the social and organizational context

Participants highlighted the crucial role of social support from colleagues in coping with MHPs. Colleagues provided opportunities to share laughter, discuss hardships, and provide emotional support, making the workday more enjoyable.

**Table 1. Participant characteristics (N = 18).**

| Characteristic | Value |
|---|---|
| Age at time of interview (range) | 20–61 years |
| Gender (*n*) | |
| Women/non-binary | 7 |
| Men | 11 |
| Ethnicity (*n*) | |
| Born outside Sweden | <5 |
| Educational level (*n*) | |
| Secondary education | 9 |
| Primary or higher education | 7 |
| Missing | 2 |
| Sector (*n*) | |
| Company 1: warehouse logistics | 7 |
| Company 2: industry | 6 |
| Company 3: retail | 5 |
| HAD score at 12-month follow-up (median, range) | |
| Depression | 4.5, 1–13 |
| Anxiety | 5, 0–18 |

Categories containing <5 participants are combined with other categories. HAD = Hospital Anxiety and Depression scale (range 0–21, where 0 is the lowest level of symptoms and 21 is the highest).

**Table 2. Themes developed during the reflexive thematic analysis of promoting and hindering factors for working while having mental health problems from the employee's perspective.**

| Theme | Promoting factors | Level[a] | Hindering factors | Level[a] |
|---|---|---|---|---|
| Influence of life stage when working while having MHPs | Developing resilience and acceptance | Individual | Developing resignation | Individual |
| Managing MHPs in the social and organizational context | Reasonable personal expectations, job crafting, leisure-time physical activity | Individual | Unable to change work situation, low job demands, physical work environment | Organizational |
| | Support from co-leagues through humor and validation; balancing social relations outside work | Group | Draining relationships, work-life balance | Group |
| | Support from the first-line manager through a trusting relationship and the first-line manager's actions | Leader | | |
| | Reasonable production goals, clear roles | Organizational | | |
| Preserving one's identity and agency when working while having MHPs | The workplace provides an arena to be productive, be seen, and be part of a whole | Organizational | Perceiving the cause to be primarily outside work, avoidance, and delay in seeking help | Individual |
| | | | Disclosure of a dilemma and experiences of exclusion | Group, leader, organizational |

[a]Refers to level in the IGLOO framework: individual, group, leader, organizational. The overarching level is not included as there were no promoting or hindering factor at this level.

This support ranged from "friendly banter" to deeper, more trusting relationships. Being able to express frustration about work or other issues and receiving recognition made it easier to cope with frustration. Supporting others through sharing one's own experiences of MHPs was also found to be highly meaningful and helpful to oneself.

> …also the group dynamics here, which we touched on a little earlier […] becomes a plus, and you get energy from it as well. From your colleagues. (Interview 12)

Taking personal responsibility for managing MHPs was discussed in different ways and involved looking inwards and setting boundaries to preserve one's resources and energy. Active strategies at work, such as listening to music, were described as changing the meaning of work tasks, making them more enjoyable and helping reduce stress. One's own attitude and how one tackled everyday life affected the appraisal of demands at work and in private life. The burden of being enough both at work and at home could result in feelings of guilt. Actively letting go of these thoughts and setting reasonable expectations for oneself could help make feelings less overwhelming and energy-draining.

Participants highlighted the benefits of engaging in physical activity as an outlet for feelings of frustration or stress, and when work itself was physically demanding. Physical activity also contributed to other positive health outcomes, such as better sleep. Striking a balance was key to sustainable habits.

> It's frustration and stuff probably. […] It's been several times like this, just "no, God, what am I really doing?" And then you get on the cross-trainer, go for half an hour and then when you get off and have a shower […] something like this: "Yes, well, we'll solve the problem tomorrow" [laughter]. (Interview 10)

MHPs could also be managed by surrounding oneself with friends and family. However, it could also mean refraining from contact with friends and family when relationships are energy-draining or destructive.

A trusting relationship between the participants and their first-line manager contributed to feeling safe and comfortable about disclosing MHPs. It was stressed that the manager did not need to understand what the participant was going through or experiencing; just being listened to without judgment was enough to feel supported. When an issue was identified, and the manager followed through in resolving it, it also strengthened the trust in the manager and supported working. Having a manager with personal experience of work tasks could legitimize the manager's understanding of issues in the workplace.

Participants felt seen, valued, and supported through the manager's actions, more than their words. This was experienced by participants as, for example, the manager's active listening, calling the participant at home and expressing a wish to get them back when on sick leave, not letting go of the challenges identified, guiding the participants in getting appropriate help, being on the floor with the employees and open to questions and chats.

> During everything that has happened, when I've my worst, and stayed at home, he has been so sweet to me, I think, and really supportive and tried to get me back to work and… It has meant a lot, actually. (Interview 11)

Scheduling could be a hindering factor to working while having MHPs. Having to work weekends and missing out on time with family created stress, as well as knowing that the burden of managing the kids' activities was shifted to their partner. Low job demands forced participants to invent work tasks. This was not voluntary active job crafting, but a forced behavior misaligned with their perception of productive work thereby evoking distressing feelings in the participants.

> The absolute hardest thing, which has been going on for quite a long time, is when there's nothing to do. For me, it's terribly hard. It's absolutely horrible to go to work and there's nothing to do […] you almost crash (Interview 9)

Being forced to work part-time also created a stressful situation. This hindering factor was discussed among employees in the retail sector. On an organizational level, having the possibility to vary work tasks was necessary if the tasks demanded high concentration, but it was also preferred because it provided some distractions and made work more interesting. Shifting focus and doing something else provided a break. Having reasonable goals for production made it possible to preserve energy. A clear role created simplicity and a sense of knowing what to do.

The warehouse logistics employees mentioned cold temperatures and noise as mentally and physically demanding in the work environment. The physical work environment posed a threat to a sustainable working life, with the potential risk of injuries because of physical strain. The prospect of injury was worrying for the participants and added to the mental load.

*Sometimes I feel really cold all day. […] I don't like those days, I can't function 100%. I feel constantly down with a lot of … many kilos of clothes on me. (Interview 3)*

**Preserving one's identity and agency when working while having MHPs**

Participants were hesitant to attribute their stress or mental strain to work, despite its impact on their ability to stay or perform at work. This delayed identification of the problem and access to support in the workplace. The participants could be caught in a cycle of avoidance, unable to change the situation. Participants described eventually being on the verge of breakdown before seeking help. Another catalyst for reaching for help was when the participant had several short absences, which alerted the first-line manager. This resulted in the disclosure of MHPs and was a starting point for making changes at work. In hindsight, participants regretted not seeking help earlier; however, admitting a need for help meant being vulnerable at work, which undermined the notion of an able worker.

*I worked until I couldn't take it anymore. I was so stressed that I... I didn't feel well. It was probably hard to kind of admit this to the boss. And I also thought it was very hard that they told me that I should take a rest as well. (Interview 14)*

Participants described situations when disclosure of MHPs led to negative consequences or fear of negative treatment, resulting in non-disclosure of personal information. Being belittled as a woman in a male dominated workplace, lack of acknowledgment of individual needs such as when colleagues and managers did not use preferred pronouns, or not being taken seriously by managers, were examples of stressful situations.

*Sometimes I feel like they're not listening to me. Maybe some of them think 'okay, she's... she doesn't know, like, she's a girl'. […] I say something like, 'you can do it like this, and it will be easier'. And I see them talking to each other as if they didn't hear me. (Interview 3)*

The workplace provided an arena to experience agency, ability and productivity. Overcoming work tasks that they or their colleagues did not think they would manage gave them energy and satisfaction. Being productive, both creating some-thing with one's hands and the physical exertion that resulted in bodily tiredness, could have benefits such as better sleep and feelings of satisfaction which in turn could help deal with stress and MHPs. Having a lot to do could also make time go faster, making the workday less strenuous, even though the work was physically demanding. Striking the balance between these two states of workflow and exhaustion was described as difficult.

*It's still nice to have a pace and be up and working and such. It's very physical, that work, you're quite... Most people there say, you're quite exhausted afterwards. (Interview 5)*

## Discussion

The aim of this study was to explore how private-sector employees at risk of sickness absence due to MHPs experienced promoting and hindering factors for working while having MHPs. Promoting and hindering factors were identified on multiple levels (Table 2), underscoring the importance of applying a systems approach when addressing issues related to MHPs in blue- and pink-collar workplaces. Promoting factors included social support from colleagues, a trusting relationship with the first-line manager, and engaging in leisure-time physical activity. Hindering factors were perceiving the cause of MHPs to be primarily outside work leading to delayed access to help, cold or noisy work environment, and a schedule mismatch. In addition, the results point to the influence of life stage and career phase in how employees with MHPs view the possibility of working and how they assess factors related to their work and their private life.

The theme, 'Managing MHPs in the social and organizational context', encompassed support from both colleagues and first-line managers; colleagues contribute to the psychosocial work climate and first-line managers provide guidance and work adaptations. Social support, including support from colleagues and first-line managers, has been identified in previous research as having a moderating effect on work stressors [27] and promoting continued work participation for employees with MHPs [28]. There were several descriptions in the data on the various roles that colleagues played in terms of supporting their ability to work while having MHPs. Colleagues could be the sole reason for enduring work, providing friendship, and playing a role in creating a positive work atmosphere. Although research on interventions targeting efforts to improve the social environment, such as peer support programs, is lacking, there is reason to believe that this may help build a mentally healthy workplace [29].

The workplace is one arena where participants could be valued, productive, and able; all of which contributed to capacity to work while having MHPs. Bringing up MHPs, however, conflicted with the notion of being a good worker. This notion was described in terms of one's own image of what a good worker is, likely to be influenced by a collective understanding at work and in society. In our study, participants described that disclosing MHPs to the first-line manager could have negative consequences, such as a feeling of being treated differently afterwards. The European Agency for Safety and Health at Work has provided guidance on how to support employees with MHPs, emphasizing the need for an inclusive environment where there is awareness of the stigma that can surround MHPs [30]. After disclosing MHPs to colleagues, the employees were generally surprised by the relief it could bring, and how colleagues responded. Colleagues who may have been seen as "hard" or "cold" could be the ones who showed the most compassion. An open climate in the workplace where it is possible to discuss MHPs could mean that these problems can be managed at an early stage.

The theme 'Preserving one's identity and agency when working while having MHPs' included the hindering factor of delayed help-seeking behavior if the perceived cause of MHPs was ascribed to the private sphere and not work. The first-line manager was not involved until symptoms had become too difficult to conceal and had started to have a greater impact at work, such as frequent short-term sickness absence. Although stigma or fear of negative treatment was not discussed among our participants in relation to the hindering factor of perceiving the cause of MHPs as related to the private sphere, it could be an underlying obstacle [31]. MHPs can have multiple causes. Regardless of the cause, managers have expressed difficulties in identifying early symptoms of MHPs [32] and managers are often made aware through self-disclosure from the employee or from colleagues [33]. If managers were able to detect MHPs that could be resolved and managed early, this could potentially promote working. Educating managers on mental health among employees has the potential to improve their support to employees experiencing MHPs [34].

The results also shine light on the demarcation between what is the responsibility of the employer and the employee. This study shows that this is not clear, at least for the employee. Raising awareness of MHPs among employees may be a good start. However, how this should be delivered to help the employee receive the organizational support available needs to be further developed. In addition, handling employees with symptoms of MHPs could also add to the burden of the manager [35]. In a review focusing on male-dominated workplaces, the authors conclude that mental health literacy and help-seeking behavior can be modified through interventions, although mechanisms are still unclear [36].

The physical work environment related to the organizational level in the IGLOO framework, (e.g., noise and cold temperatures) was identified as a hindering factor for working while having MHPs. A systematic review of small and medium-sized enterprises in the healthcare sector found that interventions targeting the physical work environment could reduce burnout and improve mental well-being among employees [37]. A scoping review focusing on sources of stress for emergency telecommunicators found that the working environment, e.g., lighting, noise or temperature, was a commonly reported source of negative impact on mental well-being [38]. Working towards a good physical work environment may therefore have an impact not only on physical health but also on mental health. The physical work environment is determined by several factors, but first and foremost the type of work that is to be performed in the work environment. Introducing changes into the work environment must consider the specific context.

In our study, work hours interfering with private life or work hours resulting in inadequate recuperation were hindering factors for working while having MHPs. A recent study on Swedish employees with common mental disorders found that flexible working hours were associated with decreased risk of sickness absence [39]. Adjusting working hours may be a preventive measure to consider, although this may be difficult for certain jobs.

Other incentives to working while having MHPs, such as economics or difficulty changing jobs due to educational level or the labor market, were mentioned in the interviews but are not included in the results. These processes may promote working in the short term, but it is not known if they could become a hindering factor in the longer term.

The results from this study support the conceptualization of presenteeism as a dynamic and adaptive behavior [6]. Although the study did not aim to assess whether this behavior was beneficial for the employees' health, whether it depleted their health resources, or how it affected performance at work, the results provide valuable insight into factors that may promote or hinder presenteeism and its potential implications for employee well-being.

## The results in relation to the JD-R model

Understanding how our results correspond to health impairment and motivational process of the JDR model could provide further knowledge on the processes that promote or hinder working while having MHPs.

The theme 'Influence of life stage on working while having MHPs' highlights how different phases in a person's private-life and career shape the perception and management of job demands. This has not been extensively researched in relation to the JD-R model. A previous study suggests that a life-course perspective, i.e., how an individual's job and personal demands and resources vary across the life-span, could be integrated with the JD-R model, because it provides insights into how job demands and resources change and influence the motivational and health-impairment processes during different career stages [40]. In our study, we found that over time, situations and individual priorities/motivation changed, which led to re-evaluation of one's work situation. Furthermore, caring for young children in early life and career stages and parents during later stages could be hindering factors for working while having MHPs. Therefore, our study supports adopting a life-course perspective in relation to the JD-R model. The theme 'Managing MHPs in the social and organizational context' included support from both colleagues and supervisors. In our study, support from colleagues, both in terms of the social culture at work and trusting relationships, was highlighted as a resource and buffer for job demands. This could be seen as an act of job crafting, as described in relation to the JD-R model [41]. The employees are not merely passive receivers of elements in the work environment, but actively shape their work environment and thereby their job demands and resources [41].

Within the theme 'Preserving one's identity and agency when working while having MHPs', the workplace was an arena for the employee to be seen and be capable. Coming to work was used as a coping strategy for maintaining function while experiencing MHPs. Conversely, the strategy of not seeking help at the workplace due to perceiving the symptoms to be related to private-life hindered working while having MHPs. Bakker and de Vries [42] suggest that different strategies can moderate either the health impairment or motivational pathways, which is corroborated by the results of our study. When the organization provides resources such as autonomy, influence, and social support, the workplace itself can become a

resource for the employee. This may lead to a positive cycle where employees feel empowered and supported, making them more likely to show up for work and seek help when needed.

It has been suggested that personal resources be added to the JD-R model. Leisure time physical activity as a means of dealing with stress or frustration was emphasized as a personal resource for working while having MHPs. Higher levels of leisure-time physical activity have been found to be associated with a lower risk of cardiovascular and all-cause mortality across occupations with different levels of physical activity [43]. Promoting physical activity in the workplace could have positive health effects, although should this be implemented during work hours, hindering factors for implementation still need to be addressed [44].

## Methodological considerations

This study has several strengths. First, the study included participants of different genders and age groups, providing varied experiences of working while having MHPs. The use of a semi-structured interview guide developed within the research group increased the credibility of the findings. During the interview, the interviewer used open probing questions and summarized statements to allow the respondent to elaborate on what had been said. The members of the research group came from a range of backgrounds and had extensive knowledge on workplace mental health and intervention development, which adds to the dependability of the analysis. All authors had previous experience in qualitative analysis. The detailed description of the study design, context, and analysis process allows the reader to assess the transferability of findings to other settings.

This study also has some limitations. Although the research team attempted to contact participants from the RCT (129 participants) through an initial phone call, only 18 participants agreed to take part in an interview. Non-response to the phone call was the most common reason for not participating. However, the sample included participants from all three companies, with diverse representation regarding gender, ages, and educational level. The relatively small sample size may raise questions about whether the dataset contained sufficient information to support our results. Although qualitative studies do not require a fixed number of participants, the concept of information power offers a useful framework for evaluating the adequacy of the sample [45]. In this study, the focused research question, the relevance of the research aim for the participants, and interpretation of the findings through the JD-R model all contribute to the trustworthiness and depth of the results.

The interviews were conducted one year after the baseline measurement, and symptoms likely changed between baseline and the interview. The interview questions were posed retrospectively to elicit participants' experiences in the year preceding the interview. Participants with more severe symptoms may find it too burdensome to participate in interviews, which may limit the transferability of our findings to employees with less severe symptoms. Additionally, most participants in the study were born in Sweden, which may limit the possibility to transfer findings to employees born outside Sweden or Europe. It is possible that other modes of data collection, such as observing participants at work, could have yielded more insight into the social context of the workplace and structures that may influence employees working while having MHPs.

Symptom severity may also influence how participants view their ability to work. Job demands may be appraised differently due to the severity of MHPs [46]. Participants with more severe symptoms may view the work as more demanding and identify fewer promoting factors for working while they had MHPs. Similarly, employees with lower symptom severity may overlook hindering factors. However, the participants in this study experienced a range of symptoms, so this has likely led to a nuanced dataset with a broad range of promoting and hindering factors for working while having MHPs.

Face-to-face interviews can provide rich content in terms of, e.g., body language or facial expressions that may be hard to capture through phone interviews. Participants in this study could choose between a phone or video-conferencing call and were encouraged to find a place and time that suited them and where they would be undisturbed. It is not known how this affected the interviews; however, phone interviews may be well suited for talking about sensitive topics [47]. We chose to conduct interviews remotely to lower the threshold for participating and make it more convenient for the participants.

Based on the many detailed descriptions provided by the participants, we believe that interviewing by phone did not affect the richness of the data.

The participants and the interviewer had minimal contact before the interview, which also may have influenced the willingness to talk about sensitive topics. The detailed descriptions of situations in the data suggest that the participants felt comfortable talking about the topic. The focus of the interviews was limited to the previous year, during which the participants were included in the RCT.

The richness of the data, combined with a rigorous coding process and interdisciplinary team reflections, enabled a nuanced understanding of working while having MHPs among employees in the private sector. Organizational contexts may vary across countries, but the alignment of our findings with previous research suggests that our results may be transferable beyond the Swedish setting. The detailed description of the setting and participants supports assessments of transferability.

### Directions for future research

Future research could focus on what organizational support is needed to detect early signs of MHPs in the workplace and how interventions can be tailored to the individual, paying particular attention to life stage and career phase. In addition, research is needed on how to foster a positive workplace culture to prevent deterioration and improve mental health.

### Concluding remarks

Employees at risk of sickness absence due to MHPs may perceive promoting and hindering factors for working while having MHPs at individual, group, leader and organizational levels. Thus, a multilevel approach should be considered when designing interventions aimed at employees working while having MHPs. Furthermore, the life-course perspective highlights that an employee's needs can vary depending on their life- and career-stages. These needs may differ but fostering an inclusive and supportive work environment could help reduce hindering factors to accessing help at work and potentially promote working while having MHPs.

### Supporting information

**S1 File. Interview guide.**
(DOCX)

### Acknowledgments

The authors would like to gratefully acknowledge the participants who took part in the study.

### Author contributions

**Conceptualization:** Anna Frantz, Iben Axén, Gunnar Bergström, Anna Finnes, Elisabeth Björk Brämberg.

**Data curation:** Anna Frantz.

**Formal analysis:** Anna Frantz, Iben Axén, Gunnar Bergström, Anna Finnes, Elisabeth Björk Brämberg.

**Funding acquisition:** Elisabeth Björk Brämberg.

**Investigation:** Anna Frantz, Iben Axén, Gunnar Bergström, Anna Finnes, Elisabeth Björk Brämberg.

**Methodology:** Anna Frantz, Iben Axén, Gunnar Bergström, Anna Finnes, Elisabeth Björk Brämberg.

**Project administration:** Anna Frantz, Elisabeth Björk Brämberg.

**Writing – original draft:** Anna Frantz.

**Writing – review & editing:** Anna Frantz, Iben Axén, Gunnar Bergström, Anna Finnes, Elisabeth Björk Brämberg.

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
