## [Decision Letter · Decision Letter 0]

10 Jul 2025

Dear Dr. Frantz,

Thank you for submitting your manuscript to PLOS ONE. After careful consideration, we feel that it has merit but does not fully meet PLOS ONE’s publication criteria as it currently stands. Therefore, we invite you to submit a revised version of the manuscript that addresses the points raised during the review process.

**ACADEMIC EDITOR:** The manuscript titled “Swedish private-sector employees’ experiences of facilitators of, and barriers to, staying at work with mental health problems: a qualitative study” presents a meaningful attempt to explore the experiences of employees with mental health problems who continue to stay at work. This is a socially and academically relevant topic, with the potential to contribute to both practical and theoretical discussions surrounding workplace mental health.

However, several important elements require revision to meet the expectations of a high-quality qualitative study.

Please submit your revised manuscript by Aug 24 2025 11:59PM. If you will need more time than this to complete your revisions, please reply to this message or contact the journal office at plosone@plos.org . A rebuttal letter that responds to each point raised by the academic editor and reviewer(s). You should upload this letter as a separate file labeled 'Response to Reviewers'.A marked-up copy of your manuscript that highlights changes made to the original version. You should upload this as a separate file labeled 'Revised Manuscript with Track Changes'.An unmarked version of your revised paper without tracked changes. You should upload this as a separate file labeled 'Manuscript'.

We look forward to receiving your revised manuscript.

Kind regards,

Yeon-Ha Kim

Academic Editor

PLOS ONE

Additional Editor Comments:

The following points should be addressed:

1. Introduction

The phrase "facilitators of, and barriers to" in the title essentially refers to identifying promoting and hindering factors. Therefore, I suggest revising the wording for clarity and conciseness.

The introduction requires clearer organization. The explanation regarding the PRIME study would be more appropriately placed in the Methods section, particularly under participant recruitment.

Although the authors provide a theoretical framework for defining "staying at work" (SAW), this section currently reads more like a conclusion rather than a theoretical positioning. The subheading used for this section should be removed, and the discussion should instead be integrated into the main flow of the introduction.

In addition, the conceptual distinction between SAW and presenteeism remains unclear and should be explicitly addressed. It is also important to more clearly articulate the rationale and unique contribution of this study within the existing body of research.

2. Methods

As a qualitative study, greater methodological rigor is needed. Qualitative research demands that the researcher maintain objectivity, demonstrate contextual understanding of participants, and ensure the trustworthiness of the findings.

First, the limited number of participants raises questions about whether the study achieved sufficient data saturation. A more cautious interpretation of the results is warranted, and the limitations should be clearly acknowledged.

Second, additional information is needed regarding the data analysis process. Specifically, how many researchers were involved in analyzing the interview data, and what procedures were followed to ensure analytical reliability?

The final sample of 18 participants was drawn from a larger pool within the PRIME study. Rather than detailing the selection process, this could be briefly summarized, with the emphasis placed on describing the final sample and its characteristics. Demographic and occupational details of participants should be included in the Results section.

3. Other Points

Abbreviations such as AFr, EBB, IA, and AFi should be written out in full upon first use.

The manuscript currently lacks information about participant characteristics, which should be clearly presented.

In summary, while the topic is timely and important, revisions are needed to improve the structure, theoretical clarity, and methodological transparency of the manuscript.

I hope you carefully reflect the reviewers’ suggestions in your revisions.

Reviewers' comments:

Reviewer's Responses to Questions

**Comments to the Author**

1. Is the manuscript technically sound, and do the data support the conclusions?

Reviewer #1: Partly

Reviewer #2: Yes

2. Has the statistical analysis been performed appropriately and rigorously?

Reviewer #1: N/A

Reviewer #2: Yes

3. Have the authors made all data underlying the findings in their manuscript fully available?

Reviewer #1: Yes

Reviewer #2: Yes

4. Is the manuscript presented in an intelligible fashion and written in standard English?

Reviewer #1: Yes

Reviewer #2: Yes

Reviewer #1: Thank you for the opportunity to review your manuscript, which addresses a timely and important topic: the experiences of private-sector employees with mental health problems in relation to staying at work (SAW). Your study contributes to the understanding of workplace mental health by focusing on facilitators and barriers from the employee perspective, particularly in blue- and pink-collar occupations, a population often underrepresented in such research.

The manuscript is well organized and clearly written. The use of reflexive thematic analysis is appropriate, and the application of the JD-R and IGLOO frameworks offers valuable potential for interpretation. However, to enhance the methodological rigor and interpretive strength of your study, the following major revisions are recommended:

Major Points

1. Sample Representativeness and Self-selection Bias

The study included 18 interview participants from a pool of 129 eligible individuals (14% response rate). While small samples are acceptable in qualitative research, this low participation rate introduces concerns about potential self-selection bias and limits the transferability of the findings. Please discuss this limitation more explicitly in the Methods and Discussion sections.

2. Participant Mental Health Status and Heterogeneity

While GHQ-12 and a predictive item on sickness absence were used for inclusion, there is limited discussion of the severity or heterogeneity of participants' mental health problems. More information or reflection on this point would provide helpful context for interpreting the findings.

3. Theoretical Integration

Although the manuscript references both the JD-R and IGLOO frameworks, their integration into the analysis appears superficial. The study would benefit greatly from a clearer mapping between themes and theoretical constructions. Consider adding a summary table or diagram to illustrate how each identified theme relates to specific elements of the frameworks.

4. Conceptual Clarity: SAW vs. Presenteeism

The terms “staying at work” and “presenteeism” are used somewhat interchangeably, yet they are conceptually distinct. Please clarify how you define and distinguish these concepts in the context of your study or justify their use as overlapping constructs.

Minor Suggestions

5. Language and Tone: The manuscript is generally written in clear and standard English. However, a few expressions may benefit from refinement for tone and clarity. For instance, consider replacing "battered" (p. XX) with a more neutral term like "exhausted" or "fatigued."

6. Use of Visual Aids: Figure 1 and Figure 2 are helpful but could benefit from additional explanation in the main text to maximize their contribution to the reader’s understanding.

7. Abbreviations: Consider providing a list of abbreviations (e.g., SAW, MHPs, JD-R) for reader reference and ensure that all terms are defined upon first use.

Conclusion

In sum, this is a promising and well-conceived manuscript that addresses a significant issue in occupational health. With revisions to strengthen the theoretical grounding, clarify key concepts, and address methodological limitations, the manuscript will be considerably improved and may be suitable for publication in PLOS ONE.

Reviewer #2: Thank you for the opportunity to review the manuscript titled “Swedish private-sector employees’ experiences of facilitators of, and barriers to, staying at work with mental health problems: a qualitative study.” The conclusions you draw are highly important for future academic debate in occupational health and represent a valuable contribution. However, there are still several points that need clarification:

Major comments

1. Adverse job stressors that negatively affect mental health have already been well established in prior research. Your study is predicated on the idea that employees with mental health problems face different factors, yet the theoretical basis for why these factors should differ needs to be reinforced. Please strengthen the Introduction with a clear rationale for why these alternative factors are expected in this population.

2. Lines 84–92 do not belong in the Introduction and should be relocated to the Methods section.

3. Isn’t “staying at work” essentially the inverse of job leave (turnover)? Please clarify how your exploration of “stay at work” differs from existing studies that examine predictors of job leave.

**Do you want your identity to be public for this peer review?** For information about this choice, including consent withdrawal, please see our Privacy Policy

Reviewer #1: No

Reviewer #2: No

---

## [Author Response · Author response to Decision Letter 1]

21 Aug 2025

Dear Editor,

Thank you for the possibility to revise and resubmit our manuscript with the title “Swedish private-sector employees’ experiences of facilitators of, and barriers to, staying at work with mental health problems: a qualitative study.”, submission ID PONE-D-25-25048. We would also like to thank you and the reviewers for your valuable comments. We believe these have contributed to improving our manuscript greatly. You will find our point-by-point response below. Changes in the manuscript are indicated with row number in the point-by-point response.

Yours sincerely,

Anna Frantz

Additional Editor Comments:

The following points should be addressed:

1. Introduction

The phrase "facilitators of, and barriers to" in the title essentially refers to identifying promoting and hindering factors. Therefore, I suggest revising the wording for clarity and conciseness.

Response: We agree with your suggestion of replacing the wording “facilitators of, and barriers to” and have replaced this with promoting and hindering factors throughout the document.

The introduction requires clearer organization. The explanation regarding the PRIME study would be more appropriately placed in the Methods section, particularly under participant recruitment.

Response: Regarding the description of the PRIME-study, we have moved this to the methods section, line 142-146.

Although the authors provide a theoretical framework for defining "staying at work" (SAW), this section currently reads more like a conclusion rather than a theoretical positioning. The subheading used for this section should be removed, and the discussion should instead be integrated into the main flow of the introduction. In addition, the conceptual distinction between SAW and presenteeism remains unclear and should be explicitly addressed.

Response: We have removed the sub-heading for the section ‘Theoretical underpinnings’ and integrated the description of theoretical frameworks into the introduction (line 67-68). With regards to the conceptual distinction between SAW and presenteeism, we find that it would fit better in the discussion, line 504-515.

It is also important to more clearly articulate the rationale and unique contribution of this study within the existing body of research.

Response: We have added a clarification of the study’s unique contribution in relation to previous studies, line 88-89 in the introduction.

2. Methods

As a qualitative study, greater methodological rigor is needed. Qualitative research demands that the researcher maintain objectivity, demonstrate contextual understanding of participants, and ensure the trustworthiness of the findings.

First, the limited number of participants raises questions about whether the study achieved sufficient data saturation. A more cautious interpretation of the results is warranted, and the limitations should be clearly acknowledged.

Response: Regarding the number of participants – we agree that it raises questions about whether enough data was collected. We have extended the discussion of limitations related to the number of participants on line 477-484. It is possible that a certain group was self-selected into this study – the additional information about the study participants included in the results section allows for a more nuanced interpretation of the transferability of the findings and the possible limitations of the study group.

We have reformulated the concluding remarks to make a more cautious interpretation of the results, line 523-532.

Second, additional information is needed regarding the data analysis process. Specifically, how many researchers were involved in analyzing the interview data, and what procedures were followed to ensure analytical reliability ?

Response: A clarification of how the authors contributed to the data analysis process, as well as a clarification of the procedure to ensure analytical reliability, is provided on lines 201, 203-207.

(The final sample of 18 participants was drawn from a larger pool within the PRIME study. Rather than detailing the selection process, this could be briefly summarized, with the emphasis placed on describing the final sample and its characteristics . Demographic and occupational details of participants should be included in the Results section.

Response: We have summarized the selection process more briefly in the methods section and removed redundant information (line 168-170). A table with participant demographics has been added to the results section (line 226).

3. Other Points

Abbreviations such as AFr, EBB, IA, and AFi should be written out in full upon first use.

Response: We have written out these abbreviations upon first use (lines 177-179) and reviewed all abbreviations throughout the manuscript.

The manuscript currently lacks information about participant characteristics, which should be clearly presented.

Response: A table with participant characteristics has been added (line 226)

Reviewer 1

Reviewer #1: Thank you for the opportunity to review your manuscript, which addresses a timely and important topic: the experiences of private-sector employees with mental health problems in relation to staying at work (SAW). Your study contributes to the understanding of workplace mental health by focusing on facilitators and barriers from the employee perspective, particularly in blue- and pink-collar occupations, a population often underrepresented in such research.　　The manuscript is well organized and clearly written. The use of reflexive thematic analysis is appropriate, and the application of the JD-R and IGLOO frameworks offers valuable potential for interpretation. However, to enhance the methodological rigor and interpretive strength of your study, the following major revisions are recommended:

Major Points

1. Sample Representativeness and Self-selection Bias

The study included 18 interview participants from a pool of 129 eligible individuals (14% response rate). While small samples are acceptable in qualitative research, this low participation rate introduces concerns about potential self-selection bias and limits the transferability of the findings . Please discuss this limitation more explicitly in the Methods and Discussion sections.

Response: Thank you for commenting on this. We have considered your suggestion and extended the discussion regarding our sample and transferability in the methodological discussion (lines 477-484).

2. Participant Mental Health Status and Heterogeneity

While GHQ-12 and a predictive item on sickness absence were used for inclusion, there is limited discussion of the severity or heterogeneity of participants' mental health problems . More information or reflection on this point would provide helpful context for interpreting the findings.

Response: Thank you for this suggestion. We have added information about symptom levels of anxiety and depression as measured with the Hospital anxiety and depression scale at 12 month follow-up corresponding to the time of the interview (Table 1, line 226). We have also extended the methodological discussion with regard to symptom severity (line 477-484), see text extract below.

“The interview was conducted one year after the baseline measurement, and symptoms have likely changed during the year between baseline and the interview. The interview questions were posed in a retrospective manner to elicit participant’s experiences from the year preceding the interview. Participants with more symptoms may find it too burdensome to participate in interviews which may limit transferability of findings to these individuals. Additionally,most participants in the study were born in Sweden, which may limit the transferability of findings to employees born outside Sweden or Europe”

3. Theoretical Integration

Although the manuscript references both the JD-R and IGLOO frameworks , their integration into the analysis appears superficial. The study would benefit greatly from a clearer mapping between themes and theoretical constructions. Consider adding a summary table or diagram to illustrate how each identified theme relates to specific elements of the frameworks.

Response: Thank you for this comment. Our intention was to use the IGLOO framework to identify on what level each promoting/hindering factor was situated on – individual level, group level, leader level, organizational level or overarching level. The JD-R is not used specifically in the data analysis, as the data analysis was inductive and not guided by a theory. The JD-R was used as a framework to discuss our results. We have extended the discussion of our results in relation to the JD-R (lines 433-443) and clarified our use of the IGLOO framework in Table 2 (line 348)

4. Conceptual Clarity: SAW vs. Presenteeism

The terms “staying at work” and “presenteeism” are used somewhat interchangeably, yet they are conceptually distinct. Please clarify how you define and distinguish these concepts in the context of your study or justify their use as overlapping constructs .

Response: We agree that there is room for improvement in the description of the concepts. We have added a clearer definition for staying at work in the introduction (line 67-68 ) After careful consideration, we have decided to move the section concerning presenteeism to the discussion to improve the clarity of the introduction. (line 505-516)

Minor Suggestions

5. Language and Tone:

The manuscript is generally written in clear and standard English. However, a few expressions may benefit from refinement for tone and clarity. For instance, consider replacing "battered" (p. XX) with a more neutral term like "exhausted" or "fatigued."

Response: We have reviewed the manuscript with consideration of tone and clarity. We have replaced the wording “battered” with exhausted (line 345).

6. Use of Visual Aids:

Figure 1 and Figure 2 are helpful but could benefit from additional explanation in the main text to maximize their contribution to the reader’s understanding.

Response: We have added additional explanation in the text for Figure 1, lines 211-216, and Figure 2, line 229.

7. Abbreviations:

Consider providing a list of abbreviations (e.g., SAW, MHPs, JD-R ) for reader reference and ensure that all terms are defined upon first use.

Response: Thank you for this suggestion! We have added a list of abbreviations at the end of the manuscript and defined terms upon first use.

Conclusion

In sum, this is a promising and well-conceived manuscript that addresses a significant issue in occupational health. With revisions to strengthen the theoretical grounding, clarify key concepts, and address methodological limitations, the manuscript will be considerably improved and may be suitable for publication in PLOS ONE.

Response: Thank you for your encouraging words.

Reviewer #2: Thank you for the opportunity to review the manuscript titled “Swedish private-sector employees’ experiences of facilitators of, and barriers to, staying at work with mental health problems: a qualitative study.” The conclusions you draw are highly important for future academic debate in occupational health and represent a valuable contribution. However, there are still several points that need clarification:

Major comments

1. Adverse job stressors that negatively affect mental health have already been well established in prior research. Your study is predicated on the idea that employees with mental health problems face different factors, yet the theoretical basis for why these factors should differ needs to be reinforced. Please strengthen the Introduction with a clear rationale for why these alternative factors are expected in this population.

Response : Thank you for your valuable comment. The rationale behind our study is the lack of knowledge on how employees with mental health problems experience what hinders or promotes stay at work, especially in blue collar sectors. We have added a clarification of the study’s unique contribution in relation to previous studies, line 88-89 in the introduction.

2. Lines 84–92 do not belong in the Introduction and should be relocated to the Methods section

Response: We have moved this section from the introduction to methods (line 142-146)

3. Isn’t “staying at work ” essentially the inverse of job leave (turnover)? Please clarify how your exploration of “stay at work” differs from existing studies that examine predictors of job leave.

Response: Thank you for raising this question. We have added a clarification to the definition of staying at work (line 67-68) and extended the discussion (lines 505-516). To the best of our knowledge, staying at work is not necessarily the inverse of job leave or turnover, as it potentially could be reported regardless of changing workplace or not.

---

## [Decision Letter · Decision Letter 1]

12 Sep 2025

Dear Dr. Frantz,

We look forward to receiving your revised manuscript.

Kind regards,

Yeon-Ha Kim

Academic Editor

PLOS ONE

**Additional Editor Comments:**

**Introduction:**

The rationale for the study is not clearly conveyed. The section would benefit from clearer organization. Currently, the introduction flows from "workplace MHP" to "studies of SAW" to "qualitative methods as an appropriate approach" and then to the "JD-R model," but the overall structure lacks coherence. In particular, the part on "studies of SAW" should be described more clearly and smoothly—at present, it is difficult to understand what the authors are aiming to communicate. I recommend emphasizing the need for the present study by identifying gaps in previous research. Furthermore, the JD-R model appears abruptly and its relevance to the research aims is unclear; please elaborate on its connection more clearly.

**Participants:**

The description of the participants should be more specific. While it is stated that the participants are first-line managers, it is not clear whether their companies belong to the production industry, warehouse logistics, or retail sector. Please provide additional information in Table 1, such as the number of employees in each company and the departments to which the participants belong. Also, please clarify how these characteristics are reflected in the findings.

**Results:**

Although Figure 2 and Table 2 present the themes, as well as promoting and hindering factors, there is no table presenting the themes, sub-themes, and meaningful statements derived through reflexive thematic analysis of the interviews. Please include such a table to enhance clarity and transparency of the analytical process.

Thank you for your efforts in revising the manuscript in accordance with the reviewers’ comments. However, the manuscript still appears to be insufficiently developed for publication at this stage.

Reviewers' comments:

Reviewer's Responses to Questions

**Comments to the Author**

Reviewer #1: All comments have been addressed

Reviewer #2: All comments have been addressed

2. Is the manuscript technically sound, and do the data support the conclusions?

Reviewer #1: Partly

Reviewer #2: Yes

3. Has the statistical analysis been performed appropriately and rigorously?

Reviewer #1: N/A

Reviewer #2: Yes

4. Have the authors made all data underlying the findings in their manuscript fully available?

Reviewer #1: No

Reviewer #2: Yes

5. Is the manuscript presented in an intelligible fashion and written in standard English?

Reviewer #1: Yes

Reviewer #2: Yes

Reviewer #1: Overall assessment

The authors have made commendable efforts to address most of the previous reviewer comments, particularly by clarifying the methods, expanding the discussion of limitations, and refining terminology. These revisions have improved the manuscript. However, several important issues remain unresolved. To strengthen the scientific rigor and conceptual clarity, I recommend further revisions before the manuscript can be considered for publication.

<major comments=" ">

1. Sample size, response rate, and transferability

The authors acknowledge the small sample and low participation rate (14%). While this is noted as a limitation, the discussion remains somewhat general. More explicit reflection on how this limited response may have influenced data saturation, representativeness, and potential self-selection bias would strengthen the trustworthiness of the findings.

2. Conceptual clarity: Staying at Work (SAW) vs. Presenteeism

The authors responded to earlier comments about the overlap between SAW and presenteeism. Nonetheless, the distinction is still insufficiently clear. Please articulate more explicitly how SAW differs from presenteeism in terms of definition, implications, and relevance for workplace interventions.

3. Theoretical integration (JD-R and IGLOO frameworks)

The manuscript now references the JD-R and IGLOO frameworks more clearly. However, their integration into the analysis remains limited. Consider providing a stronger mapping between empirical themes and theoretical constructs (e.g., a summary table or figure) to demonstrate how these frameworks meaningfully guided interpretation.

4. Heterogeneity of participants and severity of mental health problems

Although inclusion criteria are explained, there is still limited discussion of the variability in participants’ mental health conditions. This heterogeneity could significantly shape their experiences of staying at work. Further reflection on how severity of symptoms may have influenced responses is needed.

5. Analytical rigor and reflexivity

The use of reflexive thematic analysis is appropriate. Still, details about coding and reflexivity remain vague. For example, how were coder disagreements handled? In what ways was reflexivity actively applied throughout analysis? Clarifying these points would enhance methodological transparency.

<minor comments=" ">

6. Terminology and neutrality: While some terms have been adjusted, a few expressions still risk sounding value-laden. Please review wording carefully to ensure consistent neutrality.

7. Figures: Figures 1 and 2 are helpful but require more explanation in the text to clarify their relevance.

8. Abbreviations: Ensure that all abbreviations (e.g., SAW, MHPs, JD-R) are defined at first appearance in the text, even if they appear in the abbreviation list.

9. Repetition: Background details (e.g., PRIME study) remain somewhat repetitive across sections. Streamlining would improve readability.

The manuscript has improved, and the authors have addressed many earlier concerns. Nevertheless, additional clarification regarding sample limitations, conceptual distinctions, theoretical integration, and analytical rigor is still required. With these revisions, the paper has strong potential to make a meaningful contribution.</minor></major>

Reviewer #2: Thank you for revising the manuscript. I believe the manuscript has been sufficiently revised in response to the comments. I have no additional comment.

**Do you want your identity to be public for this peer review?** For information about this choice, including consent withdrawal, please see our Privacy Policy

Reviewer #1: No

Reviewer #2: No

---

## [Author Response · Author response to Decision Letter 2]

27 Oct 2025

Dear Editor,

Thank you for the possibility to further revise and resubmit our manuscript with the title “Swedish private-sector employees’ experiences of promoting and hindering factors for staying at work with mental health problems: a qualitative study”, submission ID PONE-D-25-25048. We would also like to thank you and the reviewers for your valuable comments. We believe these have contributed to improving our manuscript greatly. You will find our point-by-point response below. Changes in the manuscript are indicated with row number in the point-by-point response.

Yours sincerely,

Anna Frantz

Additional Editor Comments:

Introduction:

The rationale for the study is not clearly conveyed. The section would benefit from clearer organization. Currently, the introduction flows from "workplace MHP" to "studies of SAW" to "qualitative methods as an appropriate approach" and then to the "JD-R model," but the overall structure lacks coherence. In particular, the part on "studies of SAW" should be described more clearly and smoothly—at present, it is difficult to understand what the authors are aiming to communicate. I recommend emphasizing the need for the present study by identifying gaps in previous research. Furthermore, the JD-R model appears abruptly and its relevance to the research aims is unclear; please elaborate on its connection more clearly.

Author response:

Thank you for the helpful feedback! We have revised the introduction to improve clarity and to convey the rationale more clearly. We have also integrated the JD-R model into the introduction (lines 89-96) and the section on stay at work has been revised (lines 73-82) to make a clearer connection to the aim of the study and connect to the research gaps in previous literature.

Participants:

The description of the participants should be more specific. While it is stated that the participants are first-line managers, it is not clear whether their companies belong to the production industry, warehouse logistics, or retail sector. Please provide additional information in Table 1, such as the number of employees in each company and the departments to which the participants belong. Also, please clarify how these characteristics are reflected in the findings.

Author response:

Thank you for your valuable comments. We have updated table 1 to include the sector in which each participant belong (line 239). To preserve participant anonymity, we have not specified which department each employee worked in. Where applicable, we have added two sentences on results that were specific for sector, see lines 314-315 and 320.

Results:

Although Figure 2 and Table 2 present the themes, as well as promoting and hindering factors, there is no table presenting the themes, sub-themes, and meaningful statements derived through reflexive thematic analysis of the interviews. Please include such a table to enhance clarity and transparency of the analytical process.

Author response:

The findings were summarized in the three themes, without sub-themes. Within each theme we identified promoting and hindering factors, presented in Table 2. Figure 1 represents the analytical process with an example of moving from meaningful statements identified in the texts, codes, and themes. Figure 1 has been clarified with arrows marking the path from meaning units to themes.

Reviewers' comments:

6. Review Comments to the Author

Reviewer #1: Overall assessment

The authors have made commendable efforts to address most of the previous reviewer comments, particularly by clarifying the methods, expanding the discussion of limitations, and refining terminology. These revisions have improved the manuscript. However, several important issues remain unresolved. To strengthen the scientific rigor and conceptual clarity, I recommend further revisions before the manuscript can be considered for publication.

1. Sample size, response rate, and transferability

The authors acknowledge the small sample and low participation rate (14%). While this is noted as a limitation, the discussion remains somewhat general. More explicit reflection on how this limited response may have influenced data saturation, representativeness, and potential self-selection bias would strengthen the trustworthiness of the findings.

Author response:

Our study employed reflexive thematic analysis in line with Braun and Clarke [1]. As suggested by Braun and Clarke, information power instead of data saturation is discussed in our study, such as the narrow focus of the research question, participants relevant for the research aim, and the use of existing theoretical frameworks [1, 2]. We have elaborated on this in the limitations section (lines 520-528).

As the aim of the study was to explore promoting and hindering factors to working with mental health problems, the richness of the interviews and contextual detail provided was a way to ensure transferability of our findings. We have clarified this in the revised discussion and emphasized how these factors support the transferability of the findings to similar workplace contexts (lines 532-538).

We acknowledge the relatively low participation rate (14%) and have extended the methodological discussion addressing potential self-selection bias on lines 518-523. Importantly, our sample did not differ from the total study population in terms of key characteristics (age, gender), which suggests that self-selection may not have substantially influenced the findings. Nonetheless, we recognize that those who chose to participate may have had particular experiences or incentives, and we have reflected on this in the limitation (lines 539-545).

We hope these additions strengthen the methodological transparency and trustworthiness of the manuscript.

2. Conceptual clarity: Staying at Work (SAW) vs. Presenteeism

The authors responded to earlier comments about the overlap between SAW and presenteeism. Nonetheless, the distinction is still insufficiently clear. Please articulate more explicitly how SAW differs from presenteeism in terms of definition, implications, and relevance for workplace interventions.

Author response:

We appreciate the reviewer’s continued attention to the distinction between SAW and presenteeism. We acknowledge that our earlier explanation may not have sufficiently clarified the concerns raised regarding this issue. We have decided to use presenteeism in our study to improve clarity.

Our use of the term ‘stay at work’ was intentional, to take a more neutral stance in exploring promoting and hindering factors. In response to the reviewer’s comment, and upon further reflection, we have decided to replace the use of Stay at work to maintain a clearer focus on the aim of the study; exploring promoting and hindering factors for working with mental health problems. We believe the use of “working with MHPs” to better align with the exploratory nature of our study.

The concept of presenteeism has been researched and theorized, with two main lines of research defining presenteeism as (1) reduced productivity due to working while ill, and (2) as the behavior of attending work despite health problems . The latter has been suggested as it does not conflate predictors and outcomes. In line with Ruhle et al [3], we don’t label presenteeism per se as being either positive or negative for the employee or the organization. This decision keeps the focus on the behavior of working with MHPs including both positive and negative experiences, which is more in line with the aim of the study. Therefore, we choose to use the concept of presenteeism defined as working with ill-health [3].

We have revised the section on presenteeism in the discussion, lines 446-450.

3. Theoretical integration (JD-R and IGLOO frameworks)

The manuscript now references the JD-R and IGLOO frameworks more clearly. However, their integration into the analysis remains limited. Consider providing a stronger mapping between empirical themes and theoretical constructs (e.g., a summary table or figure) to demonstrate how these frameworks meaningfully guided interpretation.

Response to reviewers:

Thank you for your encouraging words. As our analysis was inductive, these frameworks were not used to guide the coding or theme development, but rather to support the interpretation of findings in the discussion section. This approach allowed us to remain open to participants’ experiences during the analysis without imposing predefined theoretical concepts on the data.

We recognize the value of more explicitly mapping empirical themes to theoretical constructs, and we have expanded the discussion to better articulate how the constructed themes relate to key elements of the JD-R model. We have also clarified how the IGLOO framework offers a useful lens for understanding the multi-level nature of the factors influencing SAW.

Given the inductive approach to the analysis, we opt not to include a summary table or figure including the theoretical constructs, as it may suggest a level of theoretical integration that was not present in the analytical process. Fitting the data to existing models would suit a more deductive analysis, instead we intended to use these frameworks to enrich the interpretation of the empirical results and highlight potential avenues for future research.

4. Heterogeneity of participants and severity of mental health problems

Although inclusion criteria are explained, there is still limited discussion of the variability in participants’ mental health conditions. This heterogeneity could significantly shape their experiences of staying at work. Further reflection on how severity of symptoms may have influenced responses is needed.

Author response:

Thank you for highlighting symptom variability among participants. We agree that the severity of mental health problems could shape individual’s experiences of SAW and have expanded the discussion to reflect this more explicitly (see lines 539-545).

In particular, we now address how symptom severity may influence participants’ perceptions of their work environment and available resources. As noted in the revised discussion:

“Symptom severity may also influence how participants view their ability to work. Job demands may be appraised differently due to severity of MHPs. Participants with more severe symptoms may view the work as more demanding and identify fewer promoting factors for working with MHPs. Similarly, employees with lower symptom severity may overlook hindering factors. As the participants in this study experienced a range of symptoms, this has likely led to a nuanced dataset with a broad range of promoting and hindering factors for working with MHPs.”

5. Analytical rigor and reflexivity

The use of reflexive thematic analysis is appropriate. Still, details about coding and reflexivity remain vague. For example, how were coder disagreements handled? In what ways was reflexivity actively applied throughout analysis? Clarifying these points would enhance methodological transparency.

Author response: Thank you for highlighting the need for greater clarity regarding our coding process and reflexivity. We have revised the methods section to provide a more detailed account of how these aspects were addressed during the analysis (see lines 199-202). Specifically, clarification has been added in how differences in coding was handled, and how reflexivity was enabled during the analysis process.

“Reflexivity during the analysis was facilitated in multiple ways; by notetaking as a way of keeping track of initial ideas and reflection, by having two researchers coding data, and through continuous meetings with the entire research group discussing assumptions around the data and results.”

6. Terminology and neutrality: While some terms have been adjusted, a few expressions still risk sounding value-laden. Please review wording carefully to ensure consistent neutrality.

Author response:

Thank you for your careful reading and thoughtful comment. We have reviewed the manuscript with particular attention to expressions that could be perceived as value-laden. While we fully appreciate the importance of maintaining a neutral and balanced tone, we were unable to identify specific instances where wording may unintentionally convey value judgments.

We do acknowledge that the results section includes direct quotes from participants, some of which may naturally evoke emotional or evaluative responses. These quotes were selected to illustrate themes and reflect the experiences of participants. As they are translated from the original language, we have reviewed them carefully to ensure that the translations remain faithful to the original meaning and tone. Additionally, the manuscript underwent professional language review.

7. Figures: Figures 1 and 2 are helpful but require more explanation in the text to clarify their relevance.

Author response:

Thank you for this comment. We have added an additional explanation of figure 1, lines 223-227. We have removed figure 2 after consideration.

8. Abbreviations: Ensure that all abbreviations (e.g., SAW, MHPs, JD-R) are defined at first appearance in the text, even if they appear in the abbreviation list.

Author response:

Thank you for your observation. We have carefully reviewed the manuscript regarding abbreviations. All abbreviations mentioned in your comment) are defined upon first appearance in the text, in addition to being listed in the abbreviation section. We have also defined IGLOO at first appearance and added it to the abbreviation list.

We also decided to remove the acronym PRIME, as it appeared redundant and did not add clarity to the manuscript.

9. Repetition: Background details (e.g., PRIME study) remain somewhat repetitive across sections. Streamlining would improve readability.

Author response:

We have followed your recommendation and attempted to focus all information about the context of the study (the PRIME trial) in one place in the manuscript (lines 141-163).

“Participants were recruited from a randomized controlled trial (RCT) evaluating a problem-solving intervention provided by the first-line manager at three private companies in Sweden (19). The three companies were large enterprises, covering the sectors of production industry, warehouse logistics and retail. Inclusion criteria were private sector employees aged 18-59 years; scoring with a cut-off ≥3 points on the General Health Questionnaire 12-item (Swedish version); answering ‘Yes, most likely’ or ‘Yes, quite likely’ on a question on self-rated risk of future sickness absence (i.e. ‘About your health – do you think you will receive sick leave benefits because of stress, anxiety or depression in the coming 12 months?’. The response format is a 4-point scale, ranging from ‘Yes, most likely’, ‘yes, quite likely’, ‘I’m not sure’, ‘no, probably not’); and understanding written and spoken Swedish. The cut-off of 3 on the General Health Questionnaire 12-items has been validated in the general population (20). Exclusion criteria were ongoing leave due to sickness or other reason, pregnancy, sickness absence due to a common mental disorder≥14 calendar days during the last 3 months; exposure to workplace bullying by the first-line manager; or planned long-term absence in the coming year. The participants in the study all had jobs that could not be performed outside the workplace. Ethical approval was granted from the Swedish Ethical Review Authority (reference numbers 2020-03114, 2021-01748). The study followed the ethical principles of the Declaration of Helsinki for research involving human subjects (21).”

Reviewer #2: Thank you for revising the manuscript. I believe the manuscript has been sufficiently revised in response to the comments. I have no additional comment.

Author response: Thank you for your encouraging words!

References

1. Braun V, Clarke V. To saturate or not to saturate? Questioning data saturation as a useful concept for thematic analysis and sample-size rationales. Qualitative Research in Sport Exercise and Health. 2021;13(2):201-16.

2. Malterud K, Siersma VD, Guassora AD. Sample Size in Qualitative Interview Studies: Guided by Information Power. Qualitative Health Research. 2016;26(13):1753-6

---

## [Decision Letter · Decision Letter 2]

24 Nov 2025

Dear Dr. Frantz,

Thank you for your efforts in revising the manuscript in accordance with the reviewers’ comments. However, the Introduction and Methods sections still require further strengthening. In addition, the manuscript needs overall English proofreading.

There were no conflict between the reviewers.

Please submit your revised manuscript by Jan 08 2026 11:59PM. If you will need more time than this to complete your revisions, please reply to this message or contact the journal office at plosone@plos.org . Please include the following items when submitting your revised manuscript:

We look forward to receiving your revised manuscript.

Kind regards,

Yeon-Ha Kim

Academic Editor

PLOS ONE

Journal Requirements:

**Additional Editor Comments:**

The manuscript needs overall English proofreading. For example, the phrase “for working with mental health problems” may be better expressed as “when working while having mental health problems.”

Introduction

The Introduction and the study objectives remain insufficiently clear.

As the objectives are clearly articulated in the abstract, I recommend incorporating the same level of clarity into the Introduction.

Please further develop lines 81–86 on page 5.

The concept of presenteeism appears abruptly without any supporting rationale, which makes it difficult for readers to understand.

Methods

The criteria for participant selection are unclear.

When recruiting workers experiencing MHPs, please specify which conditions are included under MHPs and clarify whether these were medically diagnosed or self‑reported.

You also mention presenteeism criteria, but additional details are needed, such as the name of the instrument used and whether criteria (e.g., at least 3 out of 12 items) were applied.

Furthermore, while the Methods section describes data collection and analysis procedures, it does not provide the main interview questions or follow‑up probes.

These should be presented for clarity.

Reviewers' comments:

Reviewer's Responses to Questions

**Comments to the Author**

Reviewer #1: All comments have been addressed

Reviewer #2: All comments have been addressed

2. Is the manuscript technically sound, and do the data support the conclusions?

Reviewer #1: Yes

Reviewer #2: Yes

3. Has the statistical analysis been performed appropriately and rigorously?

Reviewer #1: Yes

Reviewer #2: Yes

4. Have the authors made all data underlying the findings in their manuscript fully available?

Reviewer #1: Yes

Reviewer #2: Yes

5. Is the manuscript presented in an intelligible fashion and written in standard English?

Reviewer #1: Yes

Reviewer #2: Yes

Reviewer #1: Thank you for your careful revisions and for addressing the previous comments in depth. The manuscript has improved substantially in clarity, methodological transparency, and conceptual consistency. Your refinements—particularly regarding terminology, reflexive thematic analysis, and the elaboration of limitations—have strengthened the scientific rigor of the study. The manuscript is now very close to being ready for publication.

Below are a small number of minor suggestions that would further improve clarity and coherence.

Major (Minor) Points

1. Clarify the research gap more explicitly in the Introduction.

Although the Introduction has been improved, adding one or two sentences that clearly state why private-sector, low-skilled employees are underrepresented in prior qualitative research would strengthen the rationale for the study.

2. Add a brief statement linking Table 2 to the Results section.

Table 2 is useful, but including a sentence early in the Results to highlight that the table summarizes promoting and hindering factors across the themes would help guide readers.

3. Ensure conceptual consistency in the use of “presenteeism.”

Since you have moved away from “stay at work,” a brief explanation in the Introduction about why “presenteeism” is used as a neutral behavioral concept would strengthen coherence across sections.

4. Reflexive thematic analysis: further clarification.

Add one sentence explaining how coding discussions enhanced reflexivity (e.g., surfacing assumptions and broadening interpretation).

Consider adding a brief note on researcher positionality to increase transparency.

5. Clarify the classification of physical work environment factors.

A short explanatory sentence in the Discussion (e.g., that cold/noisy environments are organizational-level factors within the IGLOO framework but relate specifically to physical working conditions) would prevent confusion.

Minor Points

・Check for minor inconsistencies in terminology (e.g., “first-line manager” vs. “first line manager”).

・Consider briefly clarifying that “life-stage” in Theme 1 includes both family responsibilities and career stage.

・Conduct a final check for consistent citation formatting.

Your manuscript is strong, relevant, and well developed. The remaining issues are minor and can be addressed with modest revisions. I look forward to seeing the revised version.

Reviewer #2: Thank you for giving the opportunity to review the revised manuscripts.

I have no additional comments, same as the first round.

**Do you want your identity to be public for this peer review?** For information about this choice, including consent withdrawal, please see our Privacy Policy

Reviewer #1: No

Reviewer #2: No

---

## [Author Response · Author response to Decision Letter 3]

9 Jan 2026

Thank you for your efforts in revising the manuscript in accordance with the reviewers’ comments. However, the Introduction and Methods sections still require further strengthening. In addition, the manuscript needs overall English proofreading.

Author response: Thank you for your encouraging words! In addition to revising the manuscript, we followed your advice on English proofreading. The current version has now undergone a thorough English language review by a professional language editor.

Additional Editor Comments:

The manuscript needs overall English proofreading. For example, the phrase “for working with mental health problems” may be better expressed as “when working while having mental health problems.”

Author response: Thank you for this suggestion. The manuscript has now undergone a thorough English language review by a professional language editor. In addition to revising the wording in accordance with your suggestion above, minor changes have been made throughout the manuscript to increase clarity.

Introduction

The Introduction and the study objectives remain insufficiently clear.

As the objectives are clearly articulated in the abstract, I recommend incorporating the same level of clarity into the Introduction.

Please further develop lines 81–86 on page 5.

The concept of presenteeism appears abruptly without any supporting rationale, which makes it difficult for readers to understand.

Author response: Thank you for these suggestions. We have revised the introduction to clarify the study rationale. We have also elaborated on the presenteeism concept. We agree that it appeared abruptly and we have moved the section on presenteeism to enhance the flow in the introduction.

“Working while having MHPs can be conceptualized through presenteeism – the behavior of going to work while experiencing ill-health [4]. Previous research has studied the potential negative impact of presenteeism on work productivity and employee health [4, 5]. However, presenteeism has also been suggested to include potential functional or therapeutic aspects, such as being an individual strategy to maintain or regain function [6]. In this study, in line with this conceptualization, working while having mental health problems (presenteeism) is therefore seen as a behavior that may have negative or positive implications [4].

Methods

The criteria for participant selection are unclear.

When recruiting workers experiencing MHPs, please specify which conditions are included under MHPs and clarify whether these were medically diagnosed or self reported.

Author response: We have clarified the criteria for selection in the first paragraph under the heading “Participants and procedures”.

“Employees at risk of sickness absence due to MHPs were identified through self-reports scoring with a cut-off of ≥3 points on the General Health Questionnaire 12-item (Swedish version) validated in the general population to detect depression [24], or answering ”yes, most likely” or ”yes, quite likely” to a question on self-rated risk of sickness absence due to MHPs in the year to come (i.e. “About your health – do you think you will receive sick leave benefits because of stress, anxiety or depression in the coming 12 months?”. The response options were “Yes, most likely”,” Yes, quite likely”, “I’m not sure”, “No, probably not”. Further inclusion criteria were 18–59 years of age, ability to understand written and spoken Swedish.”

You also mention presenteeism criteria, but additional details are needed, such as the name of the instrument used and whether criteria (e.g., at least 3 out of 12 items) were applied.

Author response: Thank you for this comment. Presenteeism was included as a theoretical construct to describe the behavior of working while ill, but we did not include any instrument as selection criteria for the sample. The inclusion criteria were based on risk for sickness absence due to mental health problems operationalized as stated above. We therefore consider the sample to be at work while also having mental health problems.

Furthermore, while the Methods section describes data collection and analysis procedures, it does not provide the main interview questions or follow up probes.

These should be presented for clarity.

Author response: We agree that presenting the questions and probes within the article could add clarity. Sample questions from the interview guide have been added to the manuscript in the first paragraph under the heading “Data collection” with referral to the full interview guide as supporting information (S1 Appendix. Interview guide).

“The interview started with a broad, open question: “Tell me about your work. What does a normal working day look like?” Examples of questions to follow were “If you think about the workplace, was there anything that made it easier to stay at work?” and “Did you use any specific strategies to be able to stay at work?”. Prompts were used to encourage elaboration, e.g., “Could you give an example of a situation?” or “Could you tell me more about that?”. The full interview guide is provided in S1 Appendix.”

Reviewers' comments:

Reviewer #1: Thank you for your careful revisions and for addressing the previous comments in depth. The manuscript has improved substantially in clarity, methodological transparency, and conceptual consistency. Your refinements—particularly regarding terminology, reflexive thematic analysis, and the elaboration of limitations—have strengthened the scientific rigor of the study. The manuscript is now very close to being ready for publication.

Below are a small number of minor suggestions that would further improve clarity and coherence.

Major (Minor) Points

1. Clarify the research gap more explicitly in the Introduction.

Although the Introduction has been improved, adding one or two sentences that clearly state why private-sector, low-skilled employees are underrepresented in prior qualitative research would strengthen the rationale for the study.

Author response: Thank you for this suggestion! We have revised the introduction to clarify the rationale for the study, with the beforementioned clarification of presenteeism, and a focus on the workplace as an arena for qualitative research on hindering and promoting factors for working while having mental health problems.

“In Sweden, employers have the responsibility to systematically work to reduce risks that can lead to work-related injury or ill-health, including MHPs [17]. Hence, the workplace provides an arena for preventing sickness absence due to MHPs [18] and understanding what promotes or hinders working while having MHPs could elucidate not only individual strategies, but also factors at a group, leader, or organizational level.”

2. Add a brief statement linking Table 2 to the Results section.

Table 2 is useful, but including a sentence early in the Results to highlight that the table summarizes promoting and hindering factors across the themes would help guide readers.

Author response: We have added a sentence to the first paragraph of the Results section referencing Table 2.

“An overview of the promoting and hindering factors is presented in Table 2.”

3. Ensure conceptual consistency in the use of “presenteeism.”

Since you have moved away from “stay at work,” a brief explanation in the Introduction about why “presenteeism” is used as a neutral behavioral concept would strengthen coherence across sections.

Author response: We have clarified the use of the presenteeism concept in the introduction, at the end of the first paragraph.

“Working while having MHPs can be conceptualized through presenteeism – the behavior of going to work while experiencing ill-health [4]. Previous research has studied the potential negative impact of presenteeism on work productivity and employee health [4, 5]. However, presenteeism has also been suggested to include potential functional or therapeutic aspects, such as being an individual strategy to maintain or regain function [6]. In this study, in line with this conceptualization, working while having mental health problems (presenteeism) is therefore seen as a behavior that may have negative or positive implications [4].

4. Reflexive thematic analysis: further clarification.

Add one sentence explaining how coding discussions enhanced reflexivity (e.g., surfacing assumptions and broadening interpretation).

Author response: We have reviewed the formulation around reflexivity in the coding process and added a sentence in the second paragraph of the Methods section on how discussions enhanced reflexivity.

“To enhance reflexivity, differences in coding and assumptions on the meaning of the data were discussed between AFr and EBB. These discussions helped bring assumptions to the surface and broaden the interpretation of the data.”

Consider adding a brief note on researcher positionality to increase transparency.

Author response: To increase transparency, ‘Notes on authors’ have been added at the end of the manuscript.

Notes on authors

Anna Frantz is a PhD student at Karolinska Institutet. She is also a licensed physiotherapist. Her PhD project focuses on sickness absence due to mental health problems with both a rehabilitative perspective in a primary healthcare setting, and a secondary preventive perspective among private-sector employees.

Iben Axén, PhD, licensed chiropractor, is an Associate Professor in Musculoskeletal Health with a specific interest in back pain and musculoskeletal disorders. She also has an interest in intervention development and conduct.

Gunnar Bergström, PhD, Professor in Occupational health conducts research in areas such as stress and mental ill health in working life, the impact of sick leave on health and work performance, being at work despite health problems, and the importance of the organizational and social work environment for health and illness.

Anna Finnes, PhD, licensed Psychologist. Anna’s research includes developing and evaluating interventions for mental health problems with a specific return to work-focus.

Elisabeth Björk Brämberg, PhD, Reg. Nurse, is Associate Professor in Occupational Medicine at Karolinska Institutet and senior lecturer in Insurance Medicine at the University of Gothenburg. Her expertise covers quantitative and qualitative methods. She has extensive experience of intervention development with prevention of sick-leave and return-to-work focus.

5. Clarify the classification of physical work environment factors.

A short explanatory sentence in the Discussion (e.g., that cold/noisy environments are organizational-level factors within the IGLOO framework but relate specifically to physical working conditions) would prevent confusion.

Author response: Thank you for this suggestion. We have added a sentence referencing the organizational level of the IGLOO framework in the fourth paragraph of the discussion section.

“The physical work environment related to the organizational level in the IGLOO framework, (e.g., noise and cold temperatures) was identified as a hindering factor for working while having MHPs”

Minor Points

・Check for minor inconsistencies in terminology (e.g., “first-line manager” vs. “first line manager”).

・Consider briefly clarifying that “life-stage” in Theme 1 includes both family responsibilities and career stage.

・Conduct a final check for consistent citation formatting.

Author response: We have conducted an English language review by a professional language editor, checking for inconsistencies throughout the manuscript. We have also clarified that theme 1 includes both private- and work-life domains (in the Results section and the Discussion). We have also checked the citation formatting for consistency.

Your manuscript is strong, relevant, and well developed. The remaining issues are minor and can be addressed with modest revisions. I look forward to seeing the revised version.

Reviewer #2: Thank you for giving the opportunity to review the revised manuscripts.

I have no additional comments, same as the first round.

---

## [Decision Letter · Decision Letter 3]

28 Jan 2026

Swedish private-sector employees’ experiences of promoting and hindering factors for working while having mental health problems: a qualitative study

PONE-D-25-25048R3

Dear Dr. Frantz,

We’re pleased to inform you that your manuscript has been judged scientifically suitable for publication and will be formally accepted for publication once it meets all outstanding technical requirements.

Kind regards,

Yeon-Ha Kim

Academic Editor

PLOS One

Additional Editor Comments (optional):

Overall, the authors have adequately addressed the reviewers’ comments.

The manuscript is now well organized, with improvements made to the Introduction and Methods sections.

We appreciate the authors’ considerable efforts in revising the manuscript.

Reviewers' comments:

Reviewer's Responses to Questions

**Comments to the Author**

Reviewer #1: All comments have been addressed

2. Is the manuscript technically sound, and do the data support the conclusions?

Reviewer #1: Yes

3. Has the statistical analysis been performed appropriately and rigorously?

Reviewer #1: N/A

4. Have the authors made all data underlying the findings in their manuscript fully available?

Reviewer #1: Yes

5. Is the manuscript presented in an intelligible fashion and written in standard English?

Reviewer #1: Yes

Reviewer #1: General comments

This manuscript presents a well-conducted qualitative study exploring promoting and hindering factors for working while having mental health problems among private-sector employees. The topic is timely and highly relevant to occupational health research, particularly in relation to presenteeism and workplace mental health.

The authors have addressed the previous review comments thoroughly, and the manuscript has improved substantially in terms of conceptual clarity, methodological transparency, and overall coherence. The study aim is clearly articulated, the use of reflexive thematic analysis is appropriate, and the findings are presented in a clear and meaningful way.

The integration of a life-course perspective and the multilevel (IGLOO) framework adds conceptual depth, while remaining well aligned with the data. The discussion is balanced and adequately grounded in previous literature.

Specific comments

I have no further substantive comments. The remaining issues, if any, are minor and editorial in nature.

**Do you want your identity to be public for this peer review?** For information about this choice, including consent withdrawal, please see our Privacy Policy

Reviewer #1: No

---

## [Editor Report · Acceptance letter]

PONE-D-25-25048R3

PLOS One

Dear Dr. Frantz,

I'm pleased to inform you that your manuscript has been deemed suitable for publication in PLOS One. Congratulations! Your manuscript is now being handed over to our production team.

Kind regards,

on behalf of

Dr. Yeon-Ha Kim

Academic Editor

PLOS One